# Between-Sex Differences in Risk Factors for Cardiovascular Disease among Patients with Myocardial Infarction—A Systematic Review

**DOI:** 10.3390/jcm12155163

**Published:** 2023-08-07

**Authors:** Jack Charles Barton, Anna Wozniak, Chloe Scott, Abhisekh Chatterjee, Greg Nathan Titterton, Amber Elyse Corrigan, Ashvin Kuri, Viraj Shah, Ian Soh, Juan Carlos Kaski

**Affiliations:** 1Critical Care and Perioperative Medicine Research Group, William Harvey Research Institute, Queen Mary University of London, London E1 4NS, UK; anna.wozniak3@nhs.net (A.W.); chloe.scott19@nhs.net (C.S.); 2Department of Medicine, Faculty of Medicine, Imperial College London, London SW7 2AZ, UK; abhisekh.chatterjee20@imperial.ac.uk (A.C.); viraj.shah19@imperial.ac.uk (V.S.); 3Barts and the London School of Medicine and Dentistry, Queen Mary University of London, London E1 4NS, UK; g.n.titterton@smd20.qmul.ac.uk (G.N.T.); a.a.kuri@smd17.qmul.ac.uk (A.K.); 4Department of Medicine, Kings College London, London SE5 9RS, UK; amber.corrigan1@nhs.net; 5St. George’s University of London, London SW17 0RE, UK; ian.soh23@gmail.com; 6Molecular and Clinical Sciences Research Institute, St. George’s University of London, London SW17 0RE, UK; jkaski@sgul.ac.uk

**Keywords:** risk factors, myocardial infarction, sex, gender, women

## Abstract

Between-sex differences in the presentation, risk factors, management, and outcomes of acute myocardial infarction (MI) are well documented. However, as such differences are highly sensitive to cultural and social changes, there is a need to continuously re-evaluate the evidence. The present contemporary systematic review assesses the baseline characteristics of men and women presenting to secondary, tertiary, and quaternary centres with acute myocardial infarction (MI). Over 1.4 million participants from 18 studies, including primary prospective, cross sectional and retrospective observational studies, as well as secondary analysis of registry data are included in the study. The study showed that women were more likely than men to have a previous diagnosis of diabetes, hypertension, cerebrovascular disease, and heart failure. They also had lower odds of presenting with previous ischaemic heart disease and angina, dyslipidaemia, or a smoking history. Further work is necessary to understand the reasons for these differences, and the role that gender-specific risk factors may have in this context. Moreover, how these between-gender differences are implicated in management and outcomes also requires further work.

## 1. Introduction

Sex- and gender-bias is ubiquitous within medicine [1,2,3]. Cardiovascular medicine, and the management of myocardial infarction (MI) in particular, has long been considered a ‘disease for men’. Yet, whilst there is a higher incidence of MI in men, women tend to experience greater mortality when adjusting for age and other known confounders [4,5]. Women who survive also tend to report lower quality of life post-MI irrespective of age of presentation [6,7].

Even for the most well-evidenced therapies such as percutaneous coronary intervention (PCI), data overwhelmingly suggests that women are offered PCI less frequently, whilst experiencing lower success rates and increased complication rates [5,8,9,10]. Such differences span the entire spectrum of MI care, from acute treatment to chronic management, with women also experiencing lower rates of enrolment in cardiac rehabilitation programmes [11]. Adverse outcomes are exacerbated by ethnicity and socioeconomic status, making subsets of the population particularly vulnerable [12].

Psychosocial factors and systemic bias are also key contributors to worse outcomes in women [5,13]. Some data even suggests that adjusting for said bias all but removes the gender-outcome gap [12]. Clinical guidelines are also susceptible, given that the majority of the evidence base from which they are derived reports data collected disproportionately from men [14].

Genetic and phenotypical differences between genders explain a proportion of the differences in observed outcomes. For example, there are known gender differences in the pathophysiology of MI. Women tend to experience a greater degree of plaque erosion and embolisation, and also experience more diffuse atherosclerotic disease [15]. They are also more likely to develop microvascular and endothelial dysfunction, making their pathology less amenable to conventional therapies [15]. It follows, then, that the risk factors associated with MI in women may not be the same as those associated with MI in men. Even for shared risk factors, prevalence and associated risk are likely to vary. The same can be said for presenting clinical features [16]. In turn, these factors are likely to contribute to delayed, or incorrect diagnosis in women.

Considering the disproportionate risk faced by women, and the gender differences in the pathophysiology and presentation of MI in women, it is not enough to merely be cognisant of one’s own individual and systemic bias. We must instead evaluate the unique and dynamically changing factors which drive gender-based differences in presentation and outcomes, and integrate these findings into the evidence base underlying clinical guidelines and decision making in clinical practice. For example, whilst previous systematic reviews in this area have identified fixed risk factors in women, such as the post-menopausal loss of oestrogenic protection [17], others have identified dynamic factors that are influenced by changes to the cultural and political landscape including but not limited to physical inactivity, dietary choices, cigarette smoking, and contraceptive choices [17,18,19]. Sex-specific risk factors (which will not be addressed exhaustively in the present review) are also likely to play a major role, but are not generally included in risk assessment algorithms in clinical practice [5,15]. Greater understanding of these risk factors will inform better clinical and policy decisions that are applicable to both men and women. It could be argued that work in this field to date has contributed to the significant improvement already observed in bridging the gendered MI gap [20].

A first step in updating the literature is to consider the presence of shared, known cardiovascular risk factors between genders, and identify current knowledge gaps in the field that require further investigation. We have performed a systematic review of the last ten years of literature in order to achieve this.

## 2. Materials and Methods

The protocol for this systematic review was registered on PROSPERO (PROSPERO ID: CRD42022373892). It was conducted in accordance with the PRISMA statement [21].

### 2.1. Selection Criteria

Studies were considered if they reported risk factors by gender in those who had experienced MI either as baseline characteristics, primary, or secondary outcome. Prospective observational and retrospective observational studies of either primary or secondary data, including data originating from registries, were considered. For inclusion, the paper must have been published between January 2012 and September 2022, must have been written in English (or translated), and must have been available via institutional access.

Study population/participant inclusion criteria: any adult ≥ 16 y/o presenting to a secondary/tertiary care facility with myocardial infarction as defined by clinical evidence of acute myocardial ischaemia with a rise of cardiac high-sensitivity cardiac troponin T (>14 ng/L OR at least one value >99th percentile) or troponin I (>0.04 ng/mL at least one value >99th percentile), within 6 h of the onset of symptoms and at least one of the following: symptoms of myocardial ischaemia (cardiac chest pain); new or increased and persistent ST-segment elevation in at least two contiguous leads of ≥1 mm in all leads, other than V2-3 where elevation is ≥2.5 mm in men <40 y/o, ≥2 mm in men > 40 y/o, ≥1.5 mm in women; new horizontal or downsloping ST depression ≥ 0.5 mm in two contiguous leads and/or T wave inversion >1 m in two contiguous leads with prominent R wave or R/S ratio > 1; pathological Q waves; new or recent onset left bundle branch block; dynamic troponin T or I rise (>20% variation); regional wall motion abnormality evidenced on cardiac imaging performed within the Emergency Department [22,23]. Alternatively, studies were included if they fulfilled all other criteria and patients were deemed to have acute coronary syndrome by the senior clinician in charge of their care if the criteria for myocardial infarction was not otherwise explicitly described within the methodology, or if appropriate local or regional definition of MI was applied.

### 2.2. Literature Search

Two researchers (JB, GNT) performed an initial database search of MEDLINE, EMBASE, and CENTRAL, which was conducted between 16 and 25 November 2022. Search terms for each database can be found in Appendix A.

### 2.3. Study Selection and Data Extraction

All titles were screened by seven researchers (JB, AW, GNT, NW, CS, VS, IS), split into teams of two/three screening one database per team. If article titles were deemed to meet inclusion criteria by the two screening researchers, they were uploaded to Mendeley (Mendeley.com accessed on 20 November 2022; Mendeley Ltd., London, UK). References were then downloaded as an RIS rile and uploaded to Rayyan (rayyan.ai 4 December 2022; Rayyan Systems Inc., Cambridge, MA, USA). Duplicates were then removed.

Each article was then screened by a minimum of two researchers (all except JB, AK and JCK) to assess suitability. Where there was non-consensus between the two screening researchers, a third reviewer (JB) was sought from the research team. If the abstract was deemed to meet inclusion criteria a full paper review was then conducted, again by two researchers with a third opinion sought (primary author—JB) if there was non-consensus after initial screening.

Data extraction was undertaken at the point at which the second, or third assessor if non-consensus, deemed the study to meet inclusion criteria after full text review and was undertaken by a group of three researchers (JB, AK, AC). Information extracted included article title and DOI, citation and reference information, year of publication, study type, country/countries of data collection, number of centres, start and end date of data collection, recruitment method, participant characteristics and demographic details, prevalence of reported risk factors, inpatient mortality by sex, and additional information deemed relevant by the researcher.

### 2.4. Quality Assessment/Risk of Bias

Risk of bias assessment was undertaken if the second, or third assessor if non-consensus, deemed the study to meet inclusion criteria after full text review. One researcher not involved with initial data screening (AK) performed risk of bias assessment using the Newcastle–Ottawa Scale [24]. Each study was assessed as either high, medium, or low quality after assessment of the following domains: selection; comparability; outcome. Second opinion was sought with the lead author (JB) as deemed necessary. Outcomes of the risk of bias assessment are presented in Appendix A.

### 2.5. Statistical Analysis

Meta-analysis was not performed due to high heterogeneity between studies. Heterogeneity was attributed to a combination of primary and secondary data sources, as well as prospective and retrospective observational studies in heterogenous populations.

Fisher’s exact test and pooled odds ratios were calculated using contingency tables for all included risk factors. Alpha level was set at <5%.

## 3. Results

### 3.1. Search Results

The PRISMA flow diagram is presented in Figure 1. A total of 117 unique studies underwent full text review, 99 were excluded (see Figure 1).

### 3.2. Study Characteristics

Of the eighteen included studies, eleven were based in Europe (two in the United Kingdom [25,26], one in Germany [27], one in Iceland [28], one in Portugal [29], two in Sweden [30,31], one in Switzerland [32], one in Italy [33], one in the Netherlands [34], one in France [35]), two in the United States of America [16,36], two in India [37,38], one in Iran [39], one in Vietnam [40], and one in Australia [41]. Thirteen of the included studies were retrospective cohort studies [16,26,27,28,29,30,31,32,33,34,35,36,40], two were prospective cohort studies [25,41], two were cross-sectional studies [37,39], and one was a retrospective analysis of a prospective interventional trial [38]. In total, 580,524 women (median = 1021; IQR = 5431) and 898,800 men (median = 3220.5; IQR = 15,215.8) were included across the studies. See Table 1 for study characteristics.

### 3.3. Quality Assessment

All studies were deemed to be of high quality, with the exception of Roque et al. [29] which was deemed to be of medium quality. See the Appendix A for the full risk of bias assessment.

### 3.4. Risk Factors

Data pertaining to the odds of reported risk factors and baseline characteristics of women and men with myocardial infarction are presented in Table 2. Pooled odds ratios are displayed in Figure 2.

All of the included studies [16,25,26,27,28,29,30,31,32,33,34,35,36,37,38,39,40,41] reported prevalence of diabetes and hypertension. Women had higher odds of having both a pre-existing diagnosis of diabetes (OR 1.33; CI 1.32–1.34; *p* < 0.001) and/or a diagnosis of hypertension (OR 1.27; CI 1.26–1.28; *p* < 0.001).

A total of 16 studies reported current/ex-smoking status [16,25,26,27,28,29,30,31,33,34,35,36,37,38,39,40]. Women had lower odds of having a smoking history (OR 0.41; CI 0.41–0.41; *p* < 0.001).

A total of 12 studies reported prevalence of dyslipidaemia or hypercholesterolaemia [16,25,26,27,31,32,33,34,36,39,40,41]. Women had significantly lower odds of presenting with these diagnoses (OR 0.83; CI 0.82–0.84).

When “ischaemic heart disease”, “previous PCI”, or “previous coronary artery bypass graft” was reported by the authors [25,26,27,30,31,32,33,34,36,40] women were reported to experience significantly lower odds (OR 0.06; CI 0.063–0.065; *p* < 0.001). When authors reported anginal history without preceding MI, PCI, coronary artery bypass graft, or when authors reported anginal history independent of ischaemic heart disease [26,36,39], women experienced marginally lower odds (OR 0.95; CI 0.94–0.96; *p* < 0.001).

Women had higher odds of having a previous diagnosis of heart failure (OR 1.85; CI 1.83–1.86; *p* < 0.001), although this data was only reported by five of the included studies [26,28,32,36,39].

Three of the included studies reported prevalence of peripheral arterial disease [25,26,37], with non-significant difference observed between groups (OR 0.94; CI 0.71–1.24; *p* = 0.73).

History of stroke/cerebrovascular disease was reported in four of the included studies [28,32,33,36]. Women experienced greater odds (OR 1.57; CI 1.55–1.59; *p* < 0.001).

## 4. Discussion

The present manuscript discusses the findings of a large systematic review of 18 studies, including over 1.4 million patients. We have described the variation in prevalence of common, shared cardiac risk factors in men and women presenting with MI. Our data show that women are more likely to have pre-existing diabetes, heart failure, hypertension, and stroke prior to presenting with MI in comparison to men. By contrast, they experience lower odds of dyslipidaemia, angina and ischaemic heart disease, and smoking history. The relationships observed likely represent an amalgamation of a complex, dynamic interaction between biological, social, and cultural factors [42,43]. These factors undoubtedly impact the effectiveness of public health interventions.

Our findings largely support and update those reported by other high quality prospective observational studies [44,45,46,47]. Previous reviews have also demonstrated variation in the hazards associated with disproportionate risks, particularly those relating to diabetes, smoking, and hypertension [18,42,45]. Diabetes, for example, is known to increase the risk of MI in females by four-fold, in comparison to only two-and-a-half-fold in men [48]. By increasing the scope of our initial search and inclusion criteria, we have included studies that would not allow us to evaluate the hazards associated with the risks reported. We were thus also unable to examine the interaction between risk factors, how these interactions may be moderated by gender and sex, and what potential implications this has on patient outcomes. Continuous analysis of such data, particularly that pertaining to high quality prospective studies as well as case–control data which report data on matched controls, is necessary to prevent disproportionate bias in the future.

The included studies also failed to report variables which would have provided a more meaningful insight into the sex- and gender-differences of the most clinically relevant risk factors. Many reported diabetes mellitus in its entirety without distinguishing between types 1 and 2, or the other many subtypes of the condition. The methods of data collection employed also necessitated the use of previous allocated diagnoses, meaning that variables, particularly those relating to continuous data, could not be reported. For example, in a large retrospective cohort study of over 11,000 MI events, Rapsomaniki et al. [49] recently demonstrated the importance of distinguishing between diastolic and systolic blood pressure when evaluating one’s risk of MI. Subtle differences in likely pathophysiology may also explain some of the discordance between our findings and those of other studies which were not included. For example, Smilowitz et al. [50] reported similar rates of dyslipidaemia in men and women, with women experiencing marginally higher prevalence than men. Importantly, the authors reported lower rates of dyslipidaemia in non-obstructive coronary artery disease, a condition which is more prevalent in women and one that many of the studies included in our review failed to consider. Furthermore, Huxley et al. [18] highlighted the importance of considering the hazards associated with less prevalent risk factors in women, such as cigarette smoking. However, their findings are incongruous with other large prospective observational trials [43], suggesting that there is a need to further interrogate the risk factors in question and how they interact with themselves in addition to other confounders and colliders.

A total of 11 of the included studies reported death rate by gender [16,26,27,28,29,33,34,35,38,39,40]. However, we did not examine this data given that the included papers did not explore the well-established interaction between age, comorbidity, and treatment modality [51,52,53,54], the latter of which is particularly important, given that women experience greater complication rates [55,56] and, whilst having benefitted from advancements in technology such as drug eluding stents, remain at greater risk of suboptimal outcomes. This is in part due to suboptimal postprocedural TIMI flow grade [57]. The papers also failed to address the well-reported discrepancy between genders of non-obstructive coronary artery disease, a condition experienced far more commonly in women [58,59].

It is not unlikely that a proportion of the reported data represent discrepancy in prevalence, severity, and management of MI as well as other comorbidities and confounders [42]. This, in turn, reflects both fixed biological variables [59] as well as variables relating to the well-evidenced gender bias present within healthcare [60]. Thus, our data and that of others before us represent an important opportunity for healthcare providers. Firstly, healthcare providers may be able to better risk stratify those with clinical features of MI at the point of presentation. Pre-hospital physicians and paramedics, emergency department doctors, cardiologists, inpatient physicians, triage nurses, and all of the members of the multi-disciplinary team assessing patients must recognise the increased prevalence of diabetes, heart failure, hypertension, and stroke in women presenting with MI. They must also remain cognisant of the fact that this population are less likely to receive gold-standard treatment in a timely manner, and thus may experience worse outcomes when compared to their gender- and sex-matched counterparts. Thus, active conscious effort must be made to counter the disadvantages that women may face. It also highlights a need to consider the effect of our policies and interventions on gender and sex, and how adaptations to these may disproportionately affect women. There is clear variation in the risk factors present in women and men, and males and females, presenting with MI, only some of which can be explained by biological variation. Thus, there must be an active effort to consider these factors when targeting primary prevention and evaluating our care pathways. Further research is needed to understand how these factors interact with public health interventions, and how we can better account for them in the future. Finally, and perhaps most importantly, our paper further emphasises the need for researchers to consider the implications of gender and sex [61]. In doing so, we must understand how these are implicated by research methodology, and how we can reduce bias within our study design. In this instance, due to a need to examine the data of others, we have unavoidably conflated gender and sex for this article. This is inappropriate and represents an area of academic medicine where improvement is urgently needed. Researchers in the future should aim to make clear the difference between biological sex and gender in their data and report their findings as such, where possible. Doing so will result in more meaningful and applicable data to guide clinicians and policy makers in the future.

Our data are not without limitations. In addition to the aforementioned, we have not adjusted the odds of investigated risk factors to account for variation in known and unknown confounders, including but not limited to age, which were noted to be lower in men across all included studies. This decision was made to meet our intention to provide clinically meaningful data to front door clinicians. We have not examined the relationship between these risk factors and patient outcomes, as justified above. We have also only chosen to focus on those presenting with MI, rather than considering the relative incidence between those who do and those who do not suffer from cardiovascular disease. We have not included sex-specific risk factors in this review but are aware of the importance of those risk factors may have in specific populations. Again, this was an unavailable consequence of our search strategy meeting the primary aim of expanding our inclusion criteria to include papers reporting relevant outcome data from men and women presenting with MI, as opposed to considering large population-based studies prone to bias and often reporting data pertaining to a wider range of cardiovascular diseases.

Further research is necessary to re-evaluate and update the prevalence of risk factors for cardiovascular disease in the general population. Ideally, research would consist of high-quality meta-analyses of prospective cohort studies in addition to more frequently cited retrospective observational data, which is growing in age. There is also a need for continuous re-evaluation of all of the hazards associated with the presence of risk factors in those presenting with MI, and how these are affected by gender and sex. This will contribute to the growing body of literature which has done so for risk factors most prone to cultural variation, such as cigarette smoking [18]. In doing so, one must consider the biopsychosocial variables that are likely to moderate gender- and sex-related risk, including but not limited to disproportionate risks pertaining to socioeconomic status and health literacy [62,63]. A greater understanding of such relationships will better inform our interventions, policies, and decision making in the future.

## 5. Conclusions

There is a disparity in the distribution of shared cardiovascular risk factors between men and women presenting with MI. We must continually re-evaluate the prevalence of these risk factors to better guide our diagnostic reasoning and clinical decision making. Furthermore, there is a need for researchers to consider the implications of sex and gender on their study design, and to report data both by sex and gender, as opposed to conflating the two. Further research is needed to re-evaluate the prevalence of shared and independent risk factors for cardiovascular disease, and the subsequent development of MI, in the general population. There is also a need for continuous re-evaluation of the hazards associated with the presence of risk factors, and how these are affected by gender and sex. In doing so, one must consider the biopsychosocial variables that are likely to moderate gender- and sex-related risk.

## Figures and Tables

**Figure 1 jcm-12-05163-f001:**
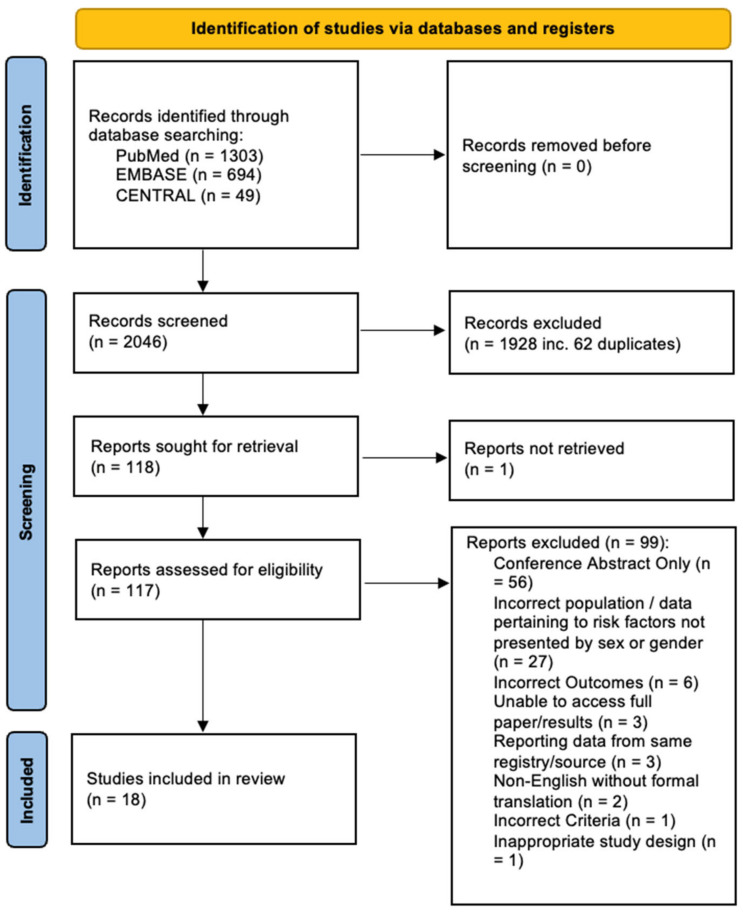
PRISMA flow diagram of the search and screening process.

**Figure 2 jcm-12-05163-f002:**
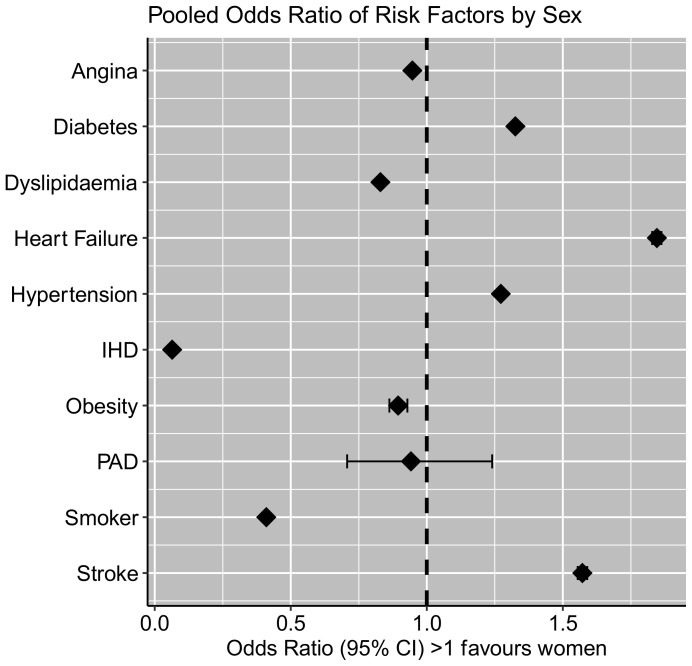
Pooled odds ratios for reported cardiovascular risk factors. Ischaemic heart disease (IHD); peripheral arterial disease (PAD). Note: ischaemic heart disease and angina reported independently as per results.

**Table 1 jcm-12-05163-t001:** Characteristics of included studies. Myocardial infarction (MI); percutaneous coronary intervention (PCI); ST-elevation myocardial infarction (STEMI).

First Author, Year of Publication	Country	Study Design	Population	Recruitment Method	Women (%)	Men (%)	Mean/Median Age (SD/IQR) Women	Mean/Median Age (SD/IQR) Men
Ahmadi et al. (2015) [39]	Iran	Cross sectional	Patients admitted with acute MI	Registry data	5717 (27.55)	15,033 (72.45)	65.4 (12.6)	59.6 (13.3)
Asleh et al. (2021) [36]	United States of America	Retrospective cohort	Patients admitted with acute MI	Retrospective electronic chart review	764 (39)	1195 (61)	73.8 (14.1)	64.2 (14)
Bajaj et al. (2016) [37]	India	Cross-sectional	Patients admitted with acute MI	Prospective recruitment of patients presenting with acute MI	50 (50)	50 (50)	62 (SD not reported)	56.5 (SD not reported)
Baumann et al. (2016) [27]	Germany	Retrospective cohort	Patients undergoing emergency PCI for acute MI	Registry data	216 (26.9)	587 (73.1)	66.8 (14.1)	60.9 (12.8)
Canto et al. (2012) [16]	United States of America	Retrospective cohort	Patients admitted with acute MI	Registry data	481,581 (42.11)	661,932 (57.89)	73.9 (12.4)	66.5 (13.2)
Dreyer et al. (2013) [41]	Australia	Prospective cohort	Patients attending PCI centre with STEMI.	Registry data	234 (25.66)	678 (74.34)	67 (14)	60 (13)
Gardarsdottir et al. (2022) [28]	Iceland	Retrospective cohort	Patients who underwent acute coronary angiography for acute MI	Retrospective analysis of prospective interventional trial dataset	625 (24.1)	1969 (75.9)	With STEMI: 68.5 (13.3) With NSTEMI: 71.0 (11.4)	With STEMI: 61.9 (12.1) With NSTEMI: 67.1 (11.6)
Khraishah et al. (2021) [38]	India	Retrospective analysis of prospective interventional trial	Patients admitted with acute MI	Prospective recruitment of all patients undergoing primary PCI at centre	5191 (24.29)	16,183 (75.71)	65 (12)	58 (12)
Krishnamurthy et al. (2019) [25]	United Kingdom	Prospective cohort	Patients undergoing primary PCI for STEMI	Retrospective electronic chart review	826 (27.09)	2223 (72.91)	69 (20)	60 (19)
Leurent et al. (2014) [35]	France	Retrospective cohort	Patients admitted with STEMI within 24 h.	Registry data	1174 (23.48)	3826 (76.52)	68.8 (14)	60.8 (12)
Nguyen et al. (2014) [40]	Vietnam	Retrospective cohort	Patients admitted with acute MI	Registry data	101 (33.44)	201 (66.56)	70 (10)	64 (12)
Ortalani et al. (2013) [33]	Italy	Retrospective cohort	Patients undergoing PCI for acute MI.	Registry data	5093 (27.75)	13,258 (72.25)	72.3 (11.2)	64.6 (12.1)
Radovanovic et al. (2012) [32]	Switzerland	Retrospective cohort	Patients presenting with STEMI.	Registry data	5786 (26.76)	15,834 (73.24)	71.5 (12.6)	62.9 (13)
Rashid et al. (2020) [26]	England and Wales	Retrospective cohort	Patients admitted with a diagnosis of NSTEMI	Registry data	40,811 (29.7%)	96,455 (70.3%)	Low Risk: 65 (55–74); Intermediate Risk: 69 (60–76); High Risk: 72 (62–79)	Low Risk: 60 (52–68); Intermediate Risk: 64 (56–72); High Risk: 66 (56–75)
Redfors et al. (2015) [31]	Sweden	Retrospective cohort	All patients treated for acute MI	Registry data	17,068 (35.47)	31,050 (64.53)	75 (12)	68 (12)
Roque et al. (2020) [29]	Portugal	Retrospective cohort	Patients admitted with acute MI	Registry data	14,177 (28.87)	34,936 (71.13)	72(12)	67(13)
Strömbäck et al. (2017) [30]	Sweden	Retrospective cohort	Patients admitted with acute MI	Registry data	242 (23.8)	775 (76.2)	61.3 (8.3)	55.8 (8.4)
Velders et al. (2013) [34]	Netherlands	Retrospective cohort	Patients who underwent primary PCI for STEMI	Prospective recruitment of all patients undergoing PCI for STEMI	868 (24.92)	2615 (75.08)	67.6 (13.1)	61.8 (11.9)

**Table 2 jcm-12-05163-t002:** Reported prevalence of baseline characteristics and risk factors of included studies. Myocardial infarction (MI); ST-elevation myocardial infarction (STEMI).

Citation	Obesity			Diabetes			Hypertension			Stroke			Dyslipidaemia/ Hypercholesterolaemia			Angina			Heart Failure			IHD/Previous Myocardial Infarction			Peripheral Artery Disease			Current/Ex Smoker		
	W	M	*p*	W	M	*p*	W	M	*p*	W	M	*p*	W	M	*p*	W	M	*p*	W	M	*p*	W	M	*p*	W	M	*p*	W	M	*p*
Ahmadi et al. (2015) [39]				33.4	18	<0.001	53.7	28.6	<0.001				25.3	15	<0.001													20.1	28.5	<0.001
Asleh et al. (2021) [36]				25.8	23.2	<0.001	79.9	62.4	<0.001				66.1	63.4	0.234													16.7	25	<0.001
Bajaj et al. (2016) [37]				52	24	0.004	46	28	0.062																			0	44	<0.001
Baumann et al. (2016) [27]				24.1	24.3	0.952	68.1	54.8	<0.001				24.1	32.5	0.021							13	18.2	0.08				35.6	55.5	<0.001
Canto et al. (2012) [16]				33.2	27		60.9	52.7		12.5	9.3		27.6	31.9		15	14.8		23.7	14.7		24	27.6					18	29	
Dreyer et al. (2013) [41]				32	27	0.24	60	45	<0.001				52	48	0.37							26	21	0.19						
Gardarsdottir et al. (2022) [28]				13	12	0.5	55	47	0.01													11	17	0.02				36	40	0.3
Khraishah et al. (2021) [38]				53.6	41.4	<0.001	61.2	42.4	<0.001																1	1	1	3.2	39.8	<0.001
Krishnamurthy et al. (2019) [25]				14.4	12.5	0.51	47.1	35.1	<0.01				30.6	30.5	0.95							10	13.5	0.02	2.1	2.8	0.43	59.6	70.4	<0.01
Leurent et al. (2014) [35]				13	11	0.06	54	36	<0.001													4	8	<0.001				26	41	<0.001
Nguyen et al. (2014) [40]				22.8	13.9		66.3	55.7					9.9	2	0.003	2	5	0.17	0	0.5	NA							1	48.3	<0.001
Ortalani et al. (2013) [33]				29.2	21.5		76.1	62		1.2	1.1		53.9	52.5					6.1	3.7		24.7	28.1					19.8	34.3	
Radovanovic et al. (2012) [32]	18.8	18.8	<0.001	22.4	17.3	<0.001	50.8	65	<0.001				49.8	53.8	<0.001							30.1	31.5	0.08						
Rashid et al. (2020) [26]				25.7	24.2		58.5	51.7					39.7	40.2		27.0	28.3		26.4	24.7					4.4	5.2		49.7	65.0	
Redfors et al. (2015) [31]				21	18	<0.001	48	38	<0.001				13	14	0.013							17	20	<0.001				21	24	<0.001
Roque et al. (2020) [29]	23.8	20.4	<0.001	26	35.9	<0.001	60.4	75.9	<0.001	8.2	6.8	<0.001							8	5	<0.001							8.5	32.6	<0.001
Strömbäck et al. (2017) [30]				33.5	20.1	<0.001	57.9	40.8	<0.001																			44.2	37.4	0.01
Velders et al. (2013) [34]				14.2	10.2	<0.001	45.9	32.5	<0.001	6.9	6.1	0.418	21.8	23.6	0.282							7.1	12.1	<0.001				40.6	47.8	0.001

## Data Availability

All reported data will be made available on request by contacting the corresponding author.

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
