# Peer review of "Between-Sex Differences in Risk Factors for Cardiovascular Disease among Patients with Myocardial Infarction—A Systematic Review"

_jcm, 2023, doi:10.3390/jcm12155163_

Round 1

Reviewer 1 Report

The present article deals with an increasingly topical issue. This topic focuses on risk factors for cardiovascular problems with gender differentiation. In general, the article can be classified as one of the better developed ones. The authors have used a number of expert sources here and the topic is focused on the chosen topic. I consider the specific focus itself to be beneficial, as there is no unnecessary extra text that would overwhelm the reader and make it difficult to navigate the text and the topic itself. The authors have focused on a systematic study, which they have handled quite well. A flow diagram is conveniently included to chart the actual selection of the studies in question. 

1. Introduction: the introduction seems generally good to me. However, somewhat weaker with respect to the paper. The authors present therapeutic approaches but miss e.g. rehabilitation effects etc. Could the authors add some articles that deal with rehabilitation for the problem at hand to complete the introduction chapter? I think the introduction is meant to offer a comprehensive view. In this case, I can recommend the articles already reviewed, which are quite successful and can help the authors to improve the chapter itself (https://www.webofscience.com/wos/woscc/full-record/WOS:000801505500001 and https://www.webofscience.com/wos/woscc/full-record/WOS:000528487500007 and https://www.webofscience.com/wos/woscc/full-record/WOS:000280339300001)

2. Materials and Methods: I rate this chapter and the subsections very highly. Here the authors have demonstrated the ability to do a very rigorous systematic study. In an indicative check, the results given fit, i.e. there is a certain logical sequence and adequacy of the procedures used. 

3. Results: a logical and engaging chapter, but I would have appreciated a better formulation and summary of the results with subsequent recommendations for practice. Such a recommendation is also not made in the discussion or conclusion. Can the authors better define their findings and recommendations? 

4-5 Discussion and conclusion: here I could imagine better wording and recommendations for practice. It would also be useful to include some coherent approaches within therapy or multidisciplinary teams.

In case the authors add to the text and make the required modifications.

Reviewer 2 Report

Barton et al did a systematic review about between sex differences in risk factors among patients with myocardial infarction including 17 mostly retrospective studies. They found that women were more likely to have a previous diagnosis of diabetes, hypertension, obesity, stroke, and heart failure, and less likely to have a history of ischemic heart disease, dyslipidemia, and smoking than men. This is an interesting and well written paper with updated information. The report follows Prisma guidelines.  The introduction gives information about the aim of the study. Myocardial infarction is defined according to current guidelines. The results are in line with previous reports. The discussion is appropriate.

I have the following comments:

Did the authors assess the degree of potential bias in each of the included study? In the supplementary appendix there is a table where studies are assessed with stars (Selection, Comparability, Outcome etc). The coding with * is not clear what this means.

I suggest, if possible, that each study might be evaluated if the risk of bias is low, high, or unclear, and that more information about the potential bias (e.g selection bias, performance bias, attrition bias reporting bias etc) is given if possible. For example: For each included study: Probability of selection bias: low, high, unclear

Probability of reporting bias: low, high, unclear

etc

Reviewer 3 Report

Clinicians can encounter sex and gender disparities in diagnostic and therapeutic responses. These disparities are noted in epidemiology, pathophysiology, clinical manifestations, disease progression, and response to treatment. The current systematic review aims to discuss the fundamental influences of sex and gender as modifiers of the traditional risk factors for CHD, and specifically for MI.

I have some comments

1. The authors cite many review articles.  Prominent citation of review articles, instead of original research papers, can obscure the connectivity of the scientific literature. Most citations, especially those focusing on previously published results, should be drawn from the original research papers. Because article citations are increasingly used as metrics of researcher productivity, the citation record affects individual scientists and their institutions. Please delete these references.

2. The issue on the controversies on sex differences in CHD mortality is almost completely ignored. The authors state” Our study also provides further evidence to support the assertion that women experience worse MI outcomes than men. However, exploring the well-established interaction between age, comorbidity, and treatment modality was considered beyond the scope of this review” This reviewer does not agree with such statement. Mortality is the most important outcome for all diseases including CHD in women.

3. The authors state that women had higher odds of having a pre-existing diagnosis of hypertension.  The authors did not cite the INTERHEART study (because the study was done in 2004). However, the findings of the INTERHEART are close to those from a recent English study of 1.25 million patients and 11, 029 MI events, in which a slightly higher relative risk of MI with increasing systolic blood pressure, but not diastolic blood pressure, was found in women compared with men (Rapsomaniki E Lancet 2014). This study should be cited and discussed as well.

4. The authors state that women had significantly lower odds of presenting with dyslipidemia or hypercholesterolemia.  The results from the Monitoring of Trends and Determinants in Cardiovascular Disease (MONICA) study reported that the increase in CHD with increasing total cholesterol holds over the entire range of patient characteristics, without any critical matter to separate risk from women and men. (Jousilahti P, Circulation 1999). The finding on dyslipidemia by the current study is also contradicted by a recent registry study (Nathaniel R. Smilowitz, Circulation: Cardiovascular Quality and Outcomes. 2017) among 322 523 patients with MI.  This study should be cited and discussed as well.

5. The authors state that women had higher odds of having a pre-existing diagnosis of diabetes. However, it is still unclear whether these observed sex differences in CHD risk are real or attributable to differences between men and women with diabetes with respect to the concomitant presence of other major risk factors for CHD or history of previous cardiovascular disease (Kania et al Archives of internal medicine 2002; Lee WL, Diabetes care 2000).

6. The authors state that women had lower odds of having a smoking history. However, “tobacco smoking represents particular harm to women and is associated with 25% higher risk of coronary heart disease compared with men.”. (Huxley RR Lancet 2011). Moreover, there are some discrepancies between studies in demonstrating a different effect of smoking as a risk factor for CHD in women compared with men. Some studies (including the INTERHEART) have shown that smoking has a similar effect on increasing the risk of CHD in both men and women (Yusuf S, Lancet 2004; Hammond EC National Cancer Institute monograph 1966;14,44,45 Tverdal A, Journal of Clinical Epidemiology 1993). Conflicting results between studies may be related to many factors including definition of smokers, age with consequent prevalence of oral contraceptive use, and synergistical action of smoking with other conventional risk factors. This issue is complicated and deserves further discussion.

7. The searching strategy did not allow, to identify some important studies which addresses the problem. For example, Manfrini et al (J Am Heart Assoc 2020) identified 14 793 patients who underwent coronary angiography for acute coronary syndromes. The main outcome measure was the association between traditional risk factors and severity of CAD. Severity of CAD was categorized as obstructive (≥50% stenosis) versus nonobstructive CAD. Current smoking and diabetes mellitus disproportionally increase the risk of obstructive CAD in women, and women with obstructive CAD had higher 30‐day mortality rates than men.

8. In figure 2 the authors show the pooled odds ratios for reported cardiovascular risk factors. CABG); IHD); PAD); and PCI. However, these data were not discussed. Some data on sex differences and disparities in outcomes after revascularization can be added in the text. Women are at higher risk of complications after coronary revascularization and other cardiac procedures, including higher rates of bleeding and additional adverse events (Poon S, Am Heart J. 2012 Lansky AJ, Circulation. 2005 Peterson PN, Circulation. 2009). Women and men have experienced larger improvements in mortality when new-generation drug- eluting stents were used. Yet the mortality reduction was proportionally similar among women and men leaving sex disparities in suboptimal postprocedural TIMI flow grade and outcomes unchanged (Cenko et al J Am Heart Assoc. 2019). Finally, women have higher complication rates following CABG and may also have higher mortality risk, (Filardo G Open Hear 2016) (Arif R, PLoS One 2016) especially in the elderly patients. The implications of these sex‐based differences that should be discussed are numerous.

Round 2

Reviewer 3 Report

Point 2, the authors did not address this comment. My concern is that presenting data on mortality is not appropriate given the authors answered that they did not explicitly explore interactions between age, comorbidity, and treatment modality with sex-dependent mortality. Therefore, they were unable to comment on these relationships based upon their findings. Accordingly, the authors should delete in the result section the following statement:” Mortality: 10 of the included studies reported death rate by gender [26, 28, 32-34, 36, 38-40]. Men were at greater odds of inpatient death (OR 1.54; CI 1.53-1.56; p<0.001)”. This statement could be misleading (page 5)

Point 3 The authors did not address this comment. They just stated that the included studies failed to report variables which would have provided a more meaningful insight into the sex- and gender-difference of the most clinically relevant risk factors. A such it is difficult to understand the scope of the current manuscript. Why did they not include other studies in the analysis?

The reviewer cited the INERHEART. The authors added such reference but did not comment on the data of this study Perhaps because the study was done in 2004. More recently, the PURE study (Walli-Attaei, Lancet 2020) replicated some of the findings of the INTERHEART and found that lower risk factor burden in women compared with men in participants with and without a history of cardiovascular disease. Overall, women had a 41% lower risk of myocardial infarction (0·59 [0·55–0·63]), a 14% lower risk of stroke (0·86 [95% CI 0·80–0·92]), a 14% lower risk of heart failure (0·86 [0·75–0·99]), and a 41% lower risk of cardiovascular death (0·59 [0·55–0·64]) compared with men. At minimum, it would have been helpful to see the authors main analyses in the subgroup with vs without a history of CVD as a sensitivity analysis.

Point 4. I see that the authors added some comments on the study by Smilowitz. The issue raised by the reviewer was that the finding on dyslipidemia by the current study (the authors state that women had significantly lower odds of presenting with dyslipidemia or hypercholesterolemia) was contradicted by many studies including that by Smilowitz. The reply was that:” Subtle differences in likely pathophysiology may also explain some of the discordance between our findings and those of other studies which were not included”. Again, why did they not include other studies (including Smilowitz) in the analysis? It is difficult to understand the scope of the current manuscript. Please clarify this.

Point 5 The reviewer complained that it is still unclear whether these observed sex differences in CHD risk are real or attributable to differences between men and women with diabetes with respect to the concomitant presence of other major risk factors for CHD or history of previous cardiovascular disease (Kania et al Archives of internal medicine 2002; Lee WL, Diabetes care 2000). The reply was that given their methodology and initial aims, they have not explored the interaction between the major CHD risk factors. Again, it is difficult to understand the scope of the current manuscript

 It appears that the authors misunderstood my comment. My comment did not relate to the fact that women had higher odds of having a pre-existing diagnosis of diabetes, but how one incorporates the study design in the analysis - my question relates to their standard errors given that the data are structured into groups constituted by different studies and centers. I would like to clarify my comment, could the authors adjust their standard errors for the clustered nature of their data (if not hospitals, then perhaps the studies listed in table 1), and if not, please explain why.

Point 6. The reviewer just required further discussion on the relationship between female sex and smoking, because the risk appears to be more pronounced in women. I got a long, convoluted explanation, but no indication of how they handled this suggestion in their manuscript. The authors response pointed out that they have expanded the discussion on cigarette smoking using the Huxley paper and the INTERHEART study. This has been used it to highlight the need to consider interactions between risk factors identified and other known and unknown variables. However, the current study did not use any interaction test as documented in figure 2 and this could be done if the authors would select an outcome.

Otherwise the take home message of the study would remain that women have higher odds of having both a pre-existing diagnosis of diabetes and/or a diagnosis of hypertension. By contrast, they have significantly lower odds of presenting with smoking history and hypercholesterolemia. A well-known story. Perhaps that the audience may be more interested to read this story is a different manner.  Indeed, the diagnosis of diabetes in a female ACS patient increases the risk of MI four-fold, whereas, in a man, it only increases the MI risk 2.5 times (Yusuf P.S. Lancet. 2004). The relative risk of an MI in a female smoker is 3.3 versus 1.9 in a male smoker (Njølstad I., Circulation. 1996) Similar impacts are seen with hypertension and obesity.

Point 7, the reviewer asked to improve the search strategy.  The searching strategy did not allow, to identify some important studies which addresses the problem. As an example, the reviewer cited a paper by   Manfrini et al (J Am Heart Assoc 2020) identified 14 793 patients who underwent coronary angiography for acute coronary syndromes. The reviewer also cited Nathaniel R. Smilowitz, Circulation: Cardiovascular Quality and Outcomes. 2017), which identified sex differences in major traditional risk factors in 322 523 patients with MI.  The authors did not include these studies in the systematic review. It should be reminded that the Selection Criteria of the study was as follows:

Studies were considered if they reported risk factors by gender in those who had experienced MI either as baseline characteristics, primary, or secondary outcome. Prospective observational and retrospective observational studies of either primary or secondary data, including data originating from registries, were considered. For inclusion, the paper must have been published between January 2012 and September 2022, must have been written in English (or translated), and must have been available via institutional access. The above-mentioned studies deserve to be included in the analysis.

Similar considerations should be done for the following studies:  Rashid M. Int. J. Cardiol. 2020:  Worrall-Carter L Women’s Health Issues. 2016; ten Haaf M.E.  Neth. Heart J. 2019; Pagidipati N.J., PLoS ONE. 2013; Shehab A., PLoS ONE. 2013: Worrall-Carter L. Cardiovasc. Interv. 2016; M Walli-Attaei, Lancet 2020:

The authors are unwilling to do so. I see that the authors commented partially in the discussion the papers of Smilowitz and Manfrini. The issue is different: why the authors did not include the above studies in the analysis? Could the authors explain this discrepancy, and/or include these studies (and the other reported above) in the analysis?

Point 8. The authors did not address most of my concerns. I would also like to note the inappropriate tone used by the authors when responding, and the statement that they could not make certain edits, for brevity and to comply with wordcount restrictions we did not do in the original manuscript. As no reply was given, I reiterate my comment as below reported.  

In figure 2 the authors show the pooled odds ratios for reported cardiovascular risk factors. CABG); IHD); PAD); and PCI. However, these data were not discussed. Some data on sex differences and disparities in outcomes after revascularization can be added in the text. Women are at higher risk of complications after coronary revascularization and other cardiac procedures, including higher rates of bleeding and additional adverse events (Poon S, Am Heart J. 2012 Lansky AJ, Circulation. 2005 Peterson PN, Circulation. 2009). Women and men have experienced larger improvements in mortality when new-generation drug- eluting stents were used. Yet the mortality reduction was proportionally similar among women and men leaving sex disparities in suboptimal postprocedural TIMI flow grade and outcomes unchanged (Cenko et al J Am Heart Assoc. 2019). Finally, women have higher complication rates following CABG and may also have higher mortality risk, (Filardo G Open Hear 2016) (Arif R, PLoS One 2016) especially in the elderly patients. The implications of these sex-based differences that should be discussed are numerous.
